# Effects of the Antidepressants Citalopram and Venlafaxine on the Big Ramshorn Snail (Planorbarius Corneus)

**Michael Ziegler [1],\*, Helene Eckstein [1], Heinz-R. Köhler [1], Selina Tisler [2], Christian Zwiener [2] and Rita Triebskorn [1,3]**

[1] Animal Physiological Ecology, University of Tübingen, Auf der Morgenstelle 5, 72076 Tübingen, Germany; eckstein.helene@yahoo.de (H.E.); heinz-r.koehler@uni-tuebingen.de (H.-R.K.); stz.oekotox@gmx.de (R.T.)

[2] Environmental Analytical Chemistry, University of Tübingen, Schnarrengerbstr. 94-96, 72076 Tübingen, Germany; seti@plen.ku.dk (S.T.); christian.zwiener@uni-tuebingen.de (C.Z.)

[3] Steinbeis Transfer Center for Ecotoxicology and Ecophysiology, Blumenstraße 13, 72108 Rottenburg, Germany

\* Correspondence: michael.ziegler@student.uni-tuebingen.de

**Abstract:** Depression is a serious health issue and, being such, treatment for it has become a topic of increasing concern. Consequently, the prescription rates of antidepressants have increased by about 50% over the past decade. Among antidepressants, citalopram and venlafaxine are the most frequently prescribed in Germany. Due to the high consumption and low elimination rates of both antidepressants during wastewater purification, they are frequently found in surface waters, where they may pose a risk to the aquatic environment. For the present study, we conducted experiments with the big ramshorn snail, which we exposed to environmentally relevant and explicitly higher concentrations (1–1000 µg/L) of the antidepressants citalopram and venlafaxine. We investigated apical endpoints, such as weight, mortality, behavioural changes, B-esterase activity, Hsp70 stress protein level and superoxide dismutase activity, as well as the tissue integrity of the hepatopancreas in the exposed snails. Citalopram and venlafaxine had no effects on the B-esterase activity, Hsp70 level and superoxide dismutase activity. Citalopram exposure resulted in weight reduction and tissue reactions in the hepatopancreas of snails exposed to 1000 µg/L. In contrast, venlafaxine did not induce comparable effects, but impacted the behaviour (sole detachment) of snails exposed to 100 µg/L and 1000 µg/L of the antidepressant. These results revealed that venlafaxine can affect snails at concentrations 10 times lower than citalopram. For this, in 2020 venlafaxine was introduced in the "Surface Water Watch List", a list of potential pollutants that should be carefully monitored in surface water by the EU Member States.

**Keywords:** antidepressant; citalopram; venlafaxine; big ramshorn snail; histopathology; behaviour; acetylcholinesterase; superoxide dismutase

## 1. Introduction

At present, depression is one of the leading non-fatal diseases with a prevalence of about 250 million cases worldwide [1]. Since 1990, the number of those diagnosed with depression has increased by 50% [2]; as a consequence, the prescription rates of antidepressants have been increasing steadily [3]. In Germany alone, the prescription rates of antidepressants increased by about 50% between 2009 and 2018. This increase can almost exclusively be explained by increased prescription rates of selective serotonin reuptake inhibitors (SSRIs) and serotonin and noradrenalin reuptake inhibitors (SNRIs) [3]. In Germany, these two drug groups accounted for 30% of the market in 2018, with the SSRI citalopram and the SNRI venlafaxine as market leaders [3]. Both substances are excreted by patients (through their urine) and are poorly eliminated in wastewater treatment plants [4–7]. Therefore, both substances enter surface waters by sewage discharge and are frequently found in water bodies in densely populated catchment areas [7–11]. Both an-

tidepressants occur, in surface waters, in the ng/L range, where venlafaxine is usually found at higher concentrations than citalopram. The MEC (measured environmental concentration) of citalopram is 219 ng/L, while the MEC of venlafaxine is 690 ng/L [8]. However, concentrations of up to 1 μg/L Venlafaxine were found in effluents of wastewater treatment plants and for citalopram concentrations up to 76 μg/L were found in Indian rivers [12–14]. For both antidepressants, adverse effects in aquatic organisms have been observed, with the majority of studies addressing behavioural effects in fish [15–23]. Regarding biochemical or histological endpoints, only a couple of studies have been conducted [24–27]. Furthermore, few data are available regarding their effects on aquatic invertebrates [26,28–32]. In this context, Buřič, et al. [29] observed decreased activity in marbled crayfish exposed to citalopram. Cuttlefish exposed to venlafaxine showed reduced camouflage ability [28]. Additionally, increased immobility in daphnids occurred after exposure to citalopram and venlafaxine, with an $EC_{50}$ of 2.74 mg/L [26,33–35]. Similarly, in gastropods, behavioural changes have been detected. Different snail species exposed to citalopram or venlafaxine have shown, for example, foot detachment from the substrate [30–32].

In the present study, we chose the great ramshorn snail *Planorbarius corneus* as our test organism. *Planorbarius corneus* is a widespread indigenous freshwater gastropod that lives in slow-flowing and stagnant European waters [36,37]. In our experiments, we exposed the snails to either venlafaxine or citalopram in concentrations between environmentally relevant 1 μg/L up to 1000 μg/L and investigated their effects on mortality, weight, behaviour, histology of the hepatopancreas, stress protein levels, and superoxide dismutase and B-esterase activities.

Behaviour is a population-relevant endpoint, which potentially affects individual fitness and, in turn, population survival [38,39]. As a behavioural endpoint in aquatic snails, the detachment of the sole from the aquarium surface has been shown to be sensitive and to reflect the impact of antidepressants as behaviour-altering substances [30–32]. In the present study, we assessed the frequency of sole detachment visually.

Furthermore, we studied the midgut gland (hepatopancreas), as the main metabolic organ in molluscs involved in digestion, as well as for the detoxification and elimination of chemicals [40,41]. Therefore, the chemicals metabolized by the hepatopancreas could have an impact on its integrity. Histological effects in the hepatopancreas of snails resulting from exposure to organic chemicals or metals have been described by several authors [42–48]. Commonly, histological effects in the hepatopancreas are accepted as indicators for the general health status of the exposed organisms. Therefore, health impairment of vital organs, like the hepatopancreas, can affect the health of an individual organism [44], which can result in effects at the population-relevant level [49–51]. Furthermore, the reproductive status of the gonads was assessed histologically, in order to distinguish effects of the reproduction status on the histopathological status of the midgut gland.

At the sub-cellular level we addressed the reactions of stress proteins and activities of the enzymes B-esterase and superoxide dismutase. The impact of venlafaxine on stress proteins in fish has been demonstrated by Maulvault, et al. [24]. As these proteins are highly conserved and act as chaperones in protein folding [52–54], the relative amount of the heat shock protein family with molecular weight of about 70 kDa (Hsp70) has been generally accepted as a biomarker for proteotoxicity, independent of the species under investigation [54]. Therefore, the Hsp70 level, as a marker for proteotoxicity, was assessed in antidepressant-exposed big ramshorn snails.

Superoxide dismutase (SOD) is one of the first-line defence enzymes protecting organisms from oxidative stress. It hydrolyses the conversion of the reactive oxygen species (ROS) superoxide anion to hydrogen peroxide [55,56]. Such ROS regularly causes oxidative stress in organisms where an imbalance of oxidants and antioxidants prevails [57,58]. Increased SOD activity has been demonstrated in daphnids and fish after exposure to several antidepressants, such as citalopram, venlafaxine, amitriptyline, or sertraline

[24,26,55,59,60]. Based on these findings, the potential induction of oxidative stress in snails exposed to citalopram and venlafaxine was assessed in the present study, through the measurement of SOD activity in the hepatopancreas.

Among others, acetylcholinesterase (AChE) and carboxylesterases (CbEs) are grouped together as B-type esterases (B-esterases), as they are all inhibited by the parathion metabolite paraxon [61,62]. AChE is an enzyme that hydrolyses the neurotransmitter acetylcholine, which is known to be inhibited by different pesticides, such as organophosphates, or pharmaceuticals, such as the anti-dementia pharmaceutical donepezil [63,64]. CbEs are able to hydrolyse carbamates and bind organophosphates and, therefore, act protectively for AChE [65–67]. Thus, the measurement of AChE activity is assumed to be a good biomarker for neuroactive compounds, including both of the considered antidepressants [55,66]. An influence of antidepressants on AChE activity has been shown for fluoxetine in common goby (*Pomatoschistus microps*) and Manila clam (*Venerupis philippinarum*) [68,69], for sertraline in goldfish [55] and for citalopram in daphnids [26]. Even though Maulvault, et al. [24] did not observe any influence of venlafaxine on the AChE activity in meagre (*Argyrosomus regius*), we assessed the effect of citalopram and venlafaxine on the activity of B-esterases in the sole and head tissue of big ramshorn snail.

These biomarkers can, therefore, act as an early warning of environmental effects precluding severe effects at the individual or population level [70].Therefore, we used multiple phenotypic traits at different biological levels (from biochemistry to the organism), in order to uncover the possible syndromic effects of both antidepressants at environmentally relevant and higher concentrations.

## 2. Material and Methods

### 2.1. Test Organism

The big ramshorn snail (*Planorbarius corneus*) is a pulmonate freshwater snail belonging to the family Planorbidae, which is native to central Europe [36,37,71]. The species is hermaphroditic, mainly feeds on algae and detritus and has a temperature tolerance between 10 and 25 °C [36,37,72,73]. The snails used in the experiments were laboratory-bred offspring of individuals obtained from Foerdefisch (Handewitt, Germany). The breeding stock was kept at room temperature in 100 L tanks filled with a mixture of filtered tap water (iron filter, active charcoal filter, particle filter) and distilled water with gravel as substrate.

### 2.2. Test Substances

Citalopram and venlafaxine were obtained from Sigma Aldrich (Steinheim, Germany), as hydrobromide and hydrochloride salts, respectively (citalopram: $C_{20}H_{21}FN_2O \cdot HBr$, CAS: 59729-32-7; venlafaxine: $C_{17}H_{27}NO_2 \cdot HCl$, CAS: 99300-78-4). Both substances were dissolved in distilled water, in order to obtain two stock solutions (100 mg/L and 1 mg/L). To receive the appropriate test solution, the respective volume of the respective stock solution was mixed with filtered tap water (iron filter, active charcoal filter, particle filter). All concentrations refer to the free base substance (citalopram: $C_{20}H_{21}FN_2O$; venlafaxine: $C_{17}H_{27}NO_2$).

### 2.3. Experimental Design

Before the experiment, the snails (>0.7 g ≈ >9 months) were transferred into an acclimation tank at room temperature, which was then placed in a climatic chamber to decrease the temperature gradually to 11 °C. This temperature was chosen because it reflects the temperatures in spring and autumn. In addition, the data were aimed to be comparable with results from earlier experiments conducted in brown trout at 11 C [60]. After one day of acclimation, the snails were transferred into the exposure tanks. The aquaria were set up in a 11 °C climatic chamber in a randomised three-block design, with

each block comprising the exposure concentrations of 0, 1, 10, 100 and 1000 µg/L of either citalopram or venlafaxine, resulting in three replicate aquaria per treatment. Concentrations were chosen to be comparable with previous publications [60,74,75]. As the number of animals of the same age was limited, we used seven snails per tank in the experiment with venlafaxine, but only five snails per tank in the experiment with citalopram. The individuals were exposed to 10 L of the respective test solutions for 29 days. Once a week, 50% of the test solution was renewed, mainly to prevent strong pollution of the tanks by faeces but also in order to obtain static exposure conditions and preserve the concentration of the substance [76]. Snails were fed, every second day, with one algae tablet (NovoPleco, JBL, Neuhofen, Deutschland) per tank. The animals were kept in a 14 h dark and 10 h light regime, and the tanks were covered with black foil to protect the snails from direct illumination. Mortality was assessed on a daily basis. During the venlafaxine experiment, the number of snails detached from the glass surface was counted daily. In the experiment with citalopram, this behaviour was not recorded. At the end of the experiment, all snails were weighed and further processed for the respective biomarker analyses. For this purpose, the shell was cracked, removed and the animals were dissected. For histopathological analyses, the gonadal part and the associated part of the hepatopancreas were chemically fixed; for the analysis of superoxide dismutase activity, B-esterase activity and for the determination of stress protein levels, the remaining hepatopancreas, head, and sole tissues, as well as the remaining visceral part of the snail, were frozen in liquid nitrogen. For both substances, five individuals per tank were sampled. The remaining two snails per venlafaxine exposure tank were frozen whole for further analyses. Water quality parameters (temperature, oxygen content, conductivity, pH) were measured at the beginning and end of the experiment.

### 2.4. Water Analysis

Water samples were taken at the beginning, after two weeks before and after water exchange and at the end of the experiment. The three replicates were pooled. Venlafaxine and citalopram water concentrations were determined by LC-MS using a 1290 Infinity HPLC system (Agilent Technologies, Waldbronn, Germany) and a triple quadrupole mass spectrometer (6490 iFunnel Triple Quadrupole LC/MS, Agilent Technologies) in ESI (+) mode. An Agilent Poroshell-120-EC-C18 (2.7 µm, 2.1 × 100 mm) column at a flow rate of 0.4 mL/min was used for separation, and the column temperature was maintained at 40 °C. Eluents A and B were water (+0.1% formic acid) and acetonitrile (+0.1% formic acid), respectively. The following gradient elution was used: 0–1 min 5% B, linear increase to 100% B within 7 min, hold for 7 min at 100% B. After switching back to the starting conditions, a reconditioning time of 3 min was employed. Samples were kept in the autosampler at 10 C with an injection volume of 1 or 10 µL (dilution factor 0–100). The detection limit of venlafaxine (mass transitions m/z 278 → 260/ → 58) and citalopram (mass transitions m/z 325 → 109/ → 262) for undiluted samples was 10 ng/L at 10 µL injection volume. A mean value of the treatment was calculated from the measured values of the sample at the beginning, in-between and at the end of the experiment. Likewise, for the physico-chemical parameters a mean value was calculated from the data of the three replicates at the beginning and the end of the experiment.

### 2.5. Behavioural Analysis

In the experiment with citalopram no abnormal behaviour was observed. In the experiment with venlafaxine, however, abnormal behaviour was detected after ≥5 days exposure to higher concentrations. Therefore, behaviour was assessed only in the venlafaxine experiment: From the fifth day of exposure onwards daily free-swimming snails in each aquarium were counted. Snails attached to the wall or bottom of the aquaria were regarded as "snails attached to the surface", whereas free-floating snails were regarded as "snails detached from the surface". Respective percentages were calculated. Detached snails, lying at the aquaria bottom, were never observed.

*2.6. Histopathology*

Samples of the hepatopancreas and the gonads were fixed in 2% glutardialdehyde dissolved in cacodylic buffer (0.01 M, pH 7.4) immediately after dissection for at least one week. Subsequently, the samples were washed three times with cacodylic buffer (0.01 M, pH 7.4), decalcified with a mixture of 98% formic acid and 70% ethanol (1:2) overnight, dehydrated in an ascending series of alcohol, and infiltrated with paraffin wax in a tissue processor (TP 1020, Leica, Wetzlar, Germany). Then, 3 μm sections were cut using a sledge microtome (SM 2000 R, Leica, Wetzlar, Germany) and mounted on glass slides. Per animal, 8 sections per slide were prepared. From each sample, one slide was stained with haematoxylin–eosin (HE), and the other with alcian blue in combination with periodic acid–Schiff reagent (alcian blue-PAS). The midgut gland samples were analysed with a light microscope (Axioskop 2, Zeiss, Oberkochen, Germany), in order to assess histopathological alterations of the tissue. First, a qualitative evaluation of the samples was conducted, followed by a semi-quantitative observer-blinded randomised evaluation. The semi-quantitative assessment was performed according to the criteria of Osterauer, et al. [44]. Samples were classified into five categories reflecting the health status: 1, control; 2, slight reaction; 3, reaction; 4, strong reaction and beginning of destruction; 5, destruction. The gonad samples were analysed, in order to assess the reproductive status of individuals.

*2.7. B-Esterase Activity*

For analysis of the B-esterase activity, the sole and head parts of the snails were used. Samples were homogenised in Tris Buffer (10 mM Tris, 10 mM NaCl) with protease inhibitors (tissue/buffer ratio 1:6 *w/v*). After centrifugation of the samples (5000 × $\vec{g}$, 10 min, 4°C), the supernatant was mixed with 10% glycerol and frozen at −20 °C until further processing. The total protein content was determined according to the method of Lowry, et al. [77] modified by Markwell, et al. [78]. Bovine serum albumin was used as a standard, and samples were photometrically measured at 650 nm. AChE activity was determined according to Ellman, et al. [79] modified by Rault, et al. [63], and measured consecutively at 405 nm for an interval of 5 min. Analysis of the activity of CbE with the substrate 4-nitrophenylacetate (NPA) was assessed according to Carr and Chambers [65] modified by Sanchez-Hernandez, et al. [67]. The activity of CbE with 4-nitrophenylvalerate (NPV) substrate was analysed according to Chanda, et al. [80] modified by Sanchez–Hernandez, et al. [67]. In both assays, the activity was measured for an interval of 5 min at 405 nm. All samples were analysed in triplicate and a mean value was calculated per sample. Each sample was treated as an independent sample, and the mean value of the treatment was calculated from all measured samples from one treatment. The enzymatic activity is expressed as milliunits per mg protein, where one unit is defined as one micromole of substrate hydrolysed per minute.

*2.8. Superoxide Dismutase Activity*

The superoxide dismutase activity (SOD) was measured in midgut gland tissue samples. The activity was determined according to the superoxide dismutase assay kit by Chayman Chemicals (Item No. 706002, Cayman Chemical Company, Ann Arbor, USA). Samples were homogenised with HEPES buffer (tissue/buffer ratio 1:5 *w/v*) and then centrifuged (1500 × $\vec{g}$, 5 min, 4 °C). Subsequently, the supernatant was taken up in Tris-HCl buffer (50 mM, pH 8; sample/buffer ratio 1:170 *v/v*) and frozen at −80 °C until further processing (no longer than 1 month). The assay was performed with duplicate samples, as previously described by Ziegler, et al. [60], and enzymatic activity measured photometrically at 450 nm. Each sample was treated as an independent sample, and the mean value of the treatment was calculated from all measured samples from one treatment. Results are given as unit/mL, where one unit is defined as the amount of enzyme needed to exhibit 50% of the superoxide radical [81].

### 2.9. Stress Protein Level

Stress protein analyses were performed with the remaining visceral part of the snail, including the gut, part of the reproductive tract, and the midgut gland. The samples were homogenised in 98% extraction buffer and 2% protease inhibitor (ratio of tissue/buffer was 1:5 *w/v*), according to the protocol of Dieterich, et al. [82]. The protein content of the samples was assessed according to Bradford [83]. To quantify the Hsp70 level, 40 µg of total protein per sample were analysed. According to Dieterich, et al. [82], protein separation was performed with sodium dodecyl sulphate polyacrylamide gel electrophoresis (SDS-PAGE), in order to separate the proteins according to their molecular weight. Subsequently, the separated proteins were western-blotted on a nitrocellulose membrane and immune-stained with a monoclonal $\alpha$-Hsp70 IgG (mouse anti-human Hsp70; Dianova, Hamburg, Germany) and secondary peroxidase-coupled $\alpha$-IgG (goat anti-mouse IgG conjugated to peroxidase; Jackson Immunoresearch, West Grove, USA). In the following, the protein bands were stained with 4-chloro-1-naphthol until they became visible. The optical volume (band area × average grey scale value) of the protein bands was quantified and referred to an internal standard (brown trout total body homogenate) [82]. Each sample was treated as an independent sample, and the mean value of the treatment was calculated from all measured samples from one treatment. Results are expressed as relative grey value.

### 2.10. Statistics

Statistical analyses were performed with the SAS JMP 14 and R 3.5.0 software (packages: lme4). Mortality was analysed using a nested Cox proportional hazards model, with replicate aquaria as the nested factor. Results for weight and biochemical biomarkers were analysed by nested ANOVA using replicate aquaria as the nested factor and a post-hoc Dunnett's test. Whenever data lacked a normal distribution, they were log-transformed. If no normal distribution could be achieved, data were evaluated with a non-parametric Kruskal–Wallis test and subsequent post-hoc Steel method with control. Detachment from the aquarium surface was analysed using a generalised linear mixed model (binomial distribution; aquarium identity as random factor) and subsequent post-hoc Dunnett's test. The alpha-level was set to 0.05. Histopathological data were analysed with the likelihood–ratio–$\chi^2$-test. If significant differences appeared, single comparisons between the control and treatment groups were conducted using the likelihood–ratio–$\chi^2$-test. To correct for multiple comparisons, the alpha level was adjusted, according to the method of Benjamini and Hochberg [84]. All depicted mean values consist of the arithmetic mean ± standard deviation.

### 2.11. CRED

Criteria for reporting and evaluating ecotoxicity data (CRED) improve the reproducibility, transparency, consistency of reliability, and evaluation of ecotoxicological data, increasing the chance that ecotoxicological studies are considered for regulatory purposes [85]. Therefore, CRED are provided in the Supplementary Materials.

## 3. Results

### 3.1. Water Parameters

The measured physico-chemical water parameters indicated suitable exposure conditions (experiment with citalopram: mean temperature 10.77 ± 0.19 °C; mean oxygen content 10.23 ± 0.19 mg/L; mean conductivity 482.98 ± 5.64 µS/cm; mean pH 8.28 ± 0.07; experiment with venlafaxine: mean temperature 12.52 ± 0.31 °C; mean oxygen content 10.08 ± 0.04 mg/L; mean conductivity 486.47 ± 3.37 µS/cm; mean pH 7.30 ± 0.20). Measured concentrations were within 79 and 140% of the nominal concentration throughout the citalopram experiment and between 70 and 88% of the nominal concentration during the venlafaxine experiment. Detailed information is provided in the Supplemental Data.

### 3.2. Mortality and Weight

In the experiment with citalopram, just a single individual died; hence, there was no effect of citalopram exposure on mortality (nested Cox proportional hazards model: $df = 4, 10$; $\chi^2 = 0$; $p = 1$). In the experiment with venlafaxine, four snails died in total, however, this was independent of the treatment. Thus, similar to the experiment with citalopram, no difference between the exposure groups and the controls could be detected (nested Cox proportional hazards model: $df = 4, 10$; $\chi^2 = 0.0632$; $p = 0.9995$).

With regard to body mass, snails exposed to citalopram were lighter than controls, which resulted in a trend of decreasing weight with increasing concentration, which became significant at 1000 μg/L citalopram (Table 1; nested ANOVA: $df = 4, 10$; $F = 2.6876$; $p = 0.0398$; post-hoc Dunnett's test: [control|1 μg/L] $p = 0.0789$ [control|10 μg/L] $p = 0.0977$ [control|100 μg/L] $p = 0.0616$ [control|1000 μg/L] $p = 0.0136$). In the experiment with venlafaxine, exposed animals in all treatment groups were slightly lighter than controls, however, these data were not significant (Table 2; nested ANOVA: $df = 4, 10$; $F = 0.6776$; $p = 0.6039$).

### 3.3. Behaviour

Snails exposed to venlafaxine presented detachment from the aquarium walls and bottom, resulting in free-floating snails at the water surface. The ratio of floaters was higher for all treatments, compared to controls; however, significant differences were only detected in snails exposed to 100 and 1000 μg/L venlafaxine (Table 2; generalised linear mixed model: $df = 4$; $F = 6.8118$; post-hoc Dunnett's test: [control|100 μg/L] $p = 0.00932$ [control|1000 μg/L] $p < 0.001$).

### 3.4. Biochemical Biomarkers

In both experiments, no significant differences between treatment and control groups were observed for any biochemical test parameter. Neither AChE activity, nor activity of both CbE endpoints were affected by the antidepressants (Tables 1 and 2; experiment with citalopram, AChE activity, nested ANOVA: $df = 4, 10$; $F = 0.2707$; $p = 0.8957$; CbE NPA activity, nested ANOVA: $df = 4, 10$; $F = 0.1822$; $p = 0.9467$; CbE NPV, nested ANOVA: $df = 4, 10$; $F = 0.0955$; experiment with venlafaxine, AChE activity, Kruskal-Wallis test: $df = 4$; $\chi^2 = 3.8627$; $p = 0.4249$; CbE NPA activity, nested ANOVA: $df = 4, 10$; $F = 2.0917$; $p = 0.093$; CbE NPV activity, nested ANOVA: $df = 4, 10$; $F = 1.7893$; $p = 0.1431$).

Superoxide dismutase activity of snails exposed to either venlafaxine or citalopram was, on average, higher than for controls. Differences in mean SOD activity amounted to up to 36% in citalopram and up to 22% in venlafaxine-exposed snails. However, no significant differences between treatment and control animals were detected (Tables 1 and 2; experiment with citalopram, nested ANOVA: $df = 4, 10$; $F = 1.8002$; $p = 0.1417$; experiment with venlafaxine, nested ANOVA: $df = 4, 10$; $F = 0.7864$; $p = 0.5385$)

Hsp70 levels of big ramshorn snails exposed to either venlafaxine or citalopram were not significantly different from those of controls (Tables 1 and 2; experiment with citalopram, nested ANOVA: $df = 4, 10$; $F = 1.124$; $p = 0.3563$; experiment with venlafaxine, nested ANOVA: $df = 4, 10$; $F = 1.4562$; $p = 0.2273$)

**Table 1.** Results obtained in the experiment with citalopram.

| Citalopram Concentration (μg/L) | 0 | 1 | 10 | 100 | 1000 |
|---|---|---|---|---|---|
| Mortality (%) | 0 ± 0 | 6.67 ± 24.49 | 0 ± 0 | 0 ± 0 | 0 ± 0 |
| Weight (g) | 2.88 ± 0.81 | 2.16 ± 0.74 | 2.19 ± 0.73 | 2.12 ± 0.90 | 1.94 ± 0.86 * |
| AChE activity (mu/mg protein) | 105.31 ± 53.79 | 113.79 ± 42.29 | 100.82 ± 30.15 | 104.33 ± 36.57 | 101.1 ± 42.02 |
| CbE NPA activity (mu/mg protein) | 149.04 ± 32.87 | 155.6 ± 27.53 | 164.09 ± 49.26 | 151.29 ± 46.89 | 156.01 ± 32.34 |
| CbE NPV activity (mu/mg protein) | 93.98 ± 44.21 | 111.73 ± 29.71 | 96.1 ± 41.32 | 79.23 ± 25.14 | 78.31 ± 29.87 |
| SOD activity (U/mL) | 96.67 ± 48.09 | 131.65 ± 50.09 | 128.87 ± 34.03 | 107.85 ± 30.64 | 105.92 ± 43.24 |

| | | | | | |
|---|---|---|---|---|---|
| Hsp70-level (relative grey value) | 0.97 ± 0.28 | 1.12 ± 0.27 | 1.18 ± 0.39 | 1.26 ± 0.34 | 1.04 ± 0.42 |
| Aqueous citalopram concentration (µg/L) | <LoD | 1.15 ± 0.04 | 8.47 ± 0.39 | 137 ± 6.48 | 1172 ± 50.41 |

Data are shown as arithmetical mean ± standard deviation. Asterisks and bold numbers represent significant differences to the respective control (* $p < 0.05$). Abbreviations: LoD, limit of detection; AChE, acetylcholinesterase; CbE, carboxylesterase; NPA, 4-nitrophenylacetate; NPV, 4-nitrophenylvalerate; SOD, superoxide dismutase, Aqueous citalopram concentration represents the arithmetic mean of the measured timepoints day 1, day 17 and day 31.

**Table 2.** Results obtained in the experiment with venlafaxine.

| Venlafaxine Concentration (µg/L) | 0 | 1 | 10 | 100 | 1000 |
|---|---|---|---|---|---|
| Mortality (%) | 4.76 ± 6.73 | 0 ± 0 | 9.52 ± 6.73 | 4.76 ± 6.73 | 0 ± 0 |
| Weight (g) | 1.93 ± 1.00 | 1.80 ± 0.63 | 1.82 ± 0.62 | 1.76 ± 0.62 | 1.52 ± 0.40 |
| Snails detached from surface (%) | 1.97 ± 3.49 | 2.54 ± 3.84 | 5 ± 4.83 | 10.16 ± 6.47 ** | 16.83 ± 6.7 *** |
| AChE activity (mu/mg Protein) | 117.10 ± 47.01 | 100.82 ± 37.95 | 89.71 ± 20.85 | 91.01 ±34.85 | 97.46 ± 21.47 |
| CbE NPA activity (mu/mg Protein) | 133.32 ± 32.92 | 151.45 ± 34.92 | 168.22 ± 23.38 | 157.35 ± 30.32 | 149.31 ± 36.36 |
| CbE NPV activity (mu/mg Protein) | 98.35 ± 52.61 | 108.20 ± 32.24 | 137.43 ± 40.56 | 111.48 ± 44.18 | 101.32 ± 36.36 |
| SOD activity (U/mL) | 114.33 ± 42.57 | 124.73 ± 38.54 | 128.01 ± 37.71 | 138.98 ± 32.21 | 127.93 ± 44.04 |
| Hsp70-level (relative grey value) | 1.39 ± 0.26 | 1.59 ± 0.37 | 1.37 ± 0.36 | 1.35 ± 0.29 | 1.34 ± 0.26 |
| Aqueous venlafaxine concentration (µg/L) | <LoD | 0.73 ± 0.03 | 8.2 ± 0.07 | 79.63 ± 3.25 | 864.15 ± 12.52 |

Data are shown as arithmetical mean ± standard deviation. Asterisks and bold numbers represent significant differences to the respective control (* $p < 0.05$; ** $p < 0.01$; *** $p < 0.001$). Abbreviations: LoD, limit of detection; AChE, acetylcholinesterase; CbE, carboxylesterase; NPA, 4-nitrophenylacetate; NPV, 4-nitrophenylvalerate; SOD, superoxide dismutase, Aqueous venlafaxine concentration represents the arithmetic mean of the measured timepoints day 1, day 20 and day 29.

*3.5. Histopathology*

The reproductive status of the gonads was assessed with respect to the spatial share of gonadal tissue containing mature oocytes and spermatozoa. Most snails showed very few mature oocytes and sperms in the gonads and, in both experiments, the reproduction status was not affected by the treatment, and no influence on the health status of the midgut gland was observed.

The hepatopancreas of snails is a tubular gland that protrudes from the midgut. Its epithelium is composed of digestive cells, crypt cells, and excretory cells [40,47]. The largest portion of the epithelium is occupied by digestive cells, which show a columnar appearance and a vacuolated cytoplasm, with vacuoles increasing in size from the apex to the cell basis (Figure 1A) [47]. Crypt cells have a large, round nucleus and are conically shaped with a wide basis (Figure 1B) [47]. Excretory cells contain large vacuoles, which often dominate the entire cell. They have been stated to derive from ageing digestive cells (Figure 1E) [86,87]. Histopathological effects which occurred in the hepatopancreas of citalopram- or venlafaxine-exposed snails were: enlargement of the tubule lumen (Figure 1C), increased hemolymph space between tubules, irregular shaped apices or bases of the tubules (Figure 1D), irregular compartmentation of the vacuoles in the digestive cells (Figure 1E), and increased number of crypt cells and irregularly shaped nuclei of the crypt cells (Figure 1F) [44].

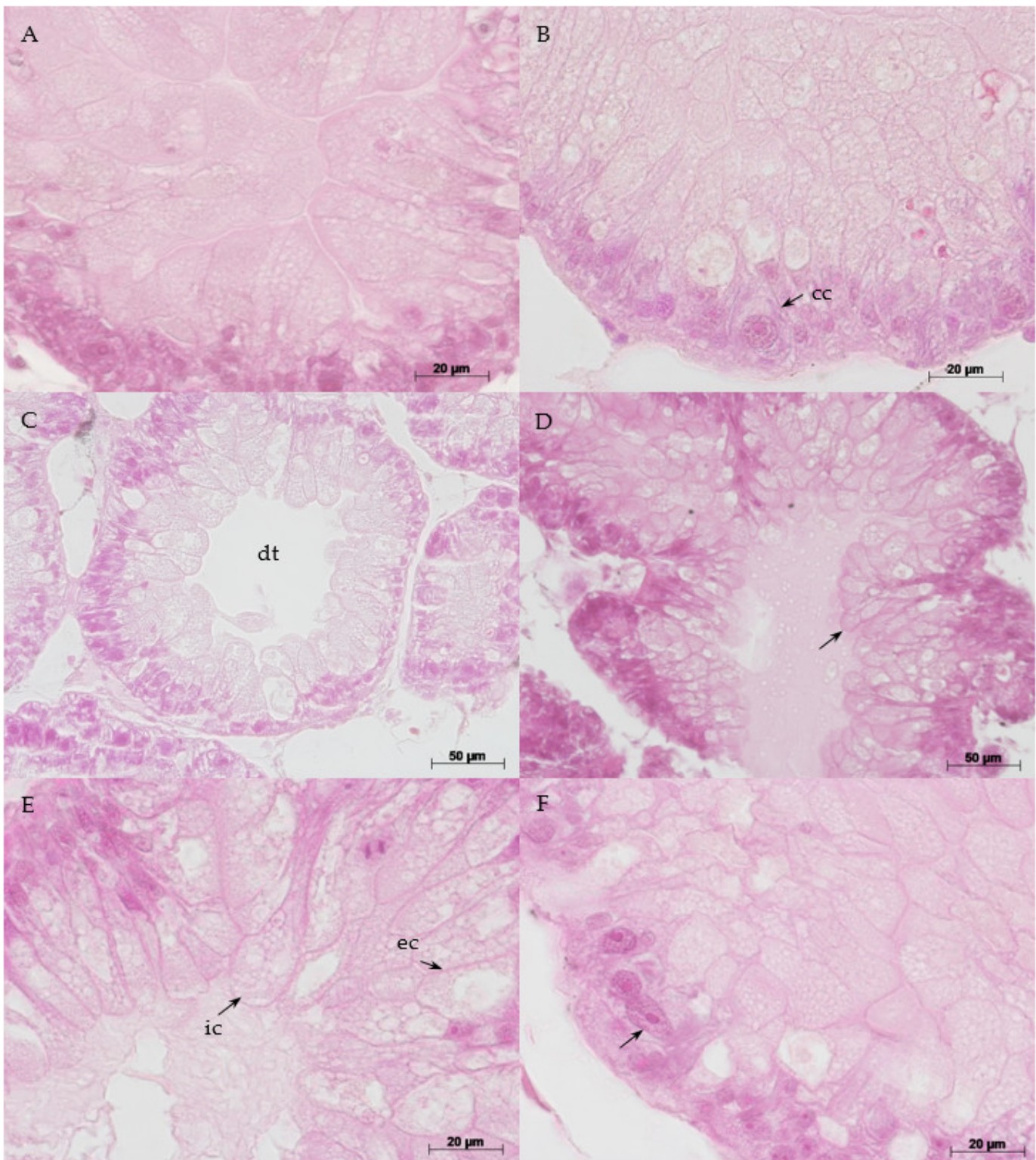

**Figure 1.** Histological status of the hepatopancreas of big ramshorn snail. All sections were stained with haematoxylin-eosin. (**A**): control status—narrow tubule lumen and digestive cells with regular compartmentation; (**B**): control status—crypt cell (cc) with round nucleus; (**C**): reaction status—dilated tubule (dt); (**D**): reaction status—slightly dilated tubule (dt) with irregular shaped apices of digestive cells and protrusion into the lumen (arrow); (**E**): reaction status—irregular compartmentation (ic) of digestive cell, excretory cell (ec); (**F**): reaction state—irregular shaped nucleus of crypt cell (arrow).

Assessment of the histopathological symptoms classified in categories 1 to 5. A: experiment with citalopram at 11 °C; B: experiment with venlafaxine at 11 °C.

Qualitative examination of the hepatopancreas of citalopram-exposed big ramshorn snails showed regular shapes of both the basis of the tubules and digestive cell nuclei in all exposure groups. Irregularly shaped apices of the tubules (Figure 1D), dilated lumina of the tubules (Figure 1C), irregular compartmentation of the digestive cells (Figure 1E), and irregular nuclei of crypt cells (Figure 1F) were detected in all treatment groups. However, the highest citalopram concentration led to more irregular shaped apices, dilated lumina of tubules, and irregular compartmentation of digestive cells, in comparison to the remaining treatment groups and the control (Supplementary Table S1). Furthermore, in all treatment groups, an increased number of crypt cells with vacuolised cytoplasm were found. Even though the semi-quantitative histopathological assessment revealed no significant differences to the control (likelihood ratio test: $df$ = 12; $\chi^2$ = 13.661; $p$ = 0.3228), there was a notable trend of impaired health conditions in snails exposed to the highest citalopram concentration (Figure 2). Detailed information of the histopathological results is given in the Supplementary Materials.

Snails exposed to venlafaxine also did not show irregular shaped bases of the tubules, or any irregular shapes of the digestive cell nuclei. In some cases, irregular-shaped apices of the tubules, dilated tubule lumina, irregular compartmentation, and irregular-shaped crypt cell nucleus were visible and distributed over all treatment groups. Furthermore, crypt cell number and vacuolised crypt cells were considerably increased in all exposure groups. The qualitative assessment of the hepatopancreas did not reveal any differences between the control and the venlafaxine-exposed animals. Similarly, the semi-quantitative assessment of the midgut gland of snails in venlafaxine experiment did not reveal any differences between control and venlafaxine exposure groups (likelihood ratio test: $df$ = 12; $\chi^2$ = 14.602; $p$ = 0.2639). For venlafaxine, detailed information on the histopathology of the hepatopancreas is also provided in the Supplementary Materials.

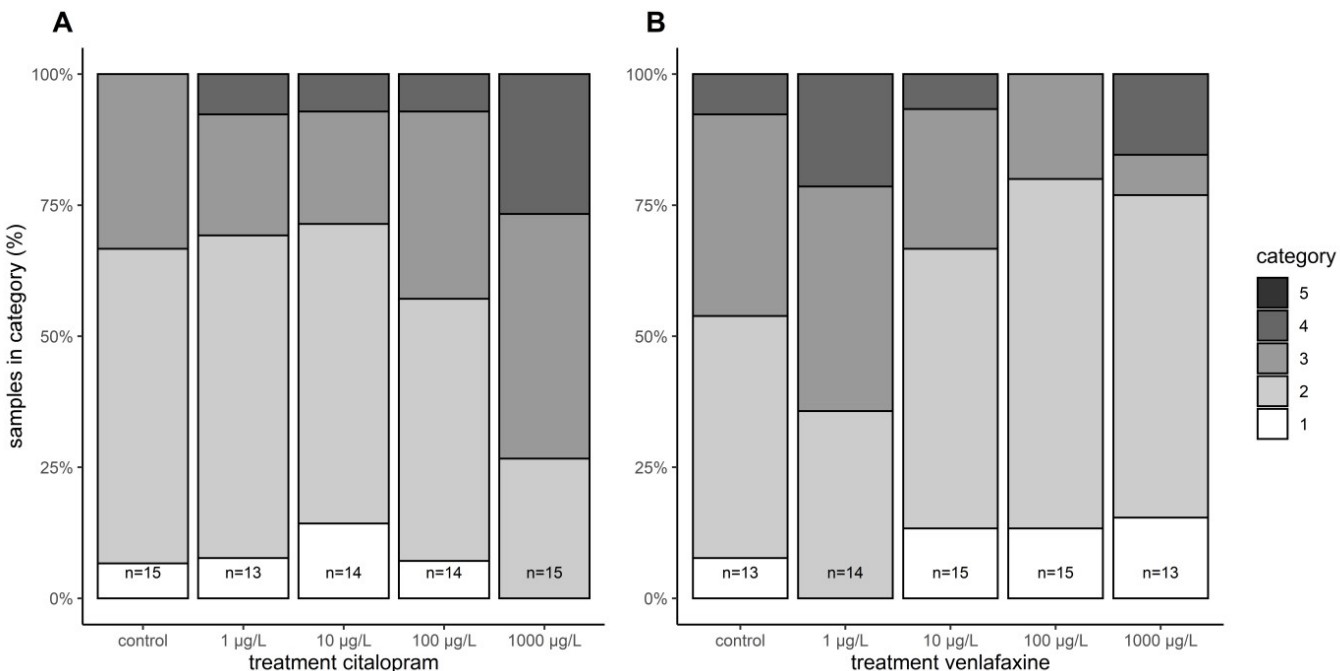

**Figure 2.** Semi-quantitative assessment of the hepatopancreas status of big ramshorn snail. A: experiment with citalopram, B experiment with venlafaxine.

## 4. Discussion

Chemical analyses showed that the real concentrations in both experiments were close to the nominal concentrations (Tables 1 and 2). The mortality of snails exposed to either venlafaxine or citalopram did not differ significantly from the control. Likewise, no influence on the mortality rate of antidepressants could be shown in Manila clam (*Venerupis philippinarum*) [69], European valve snail (*Valvata piscinalis*), New Zealand mud snail (*Potamopyrgus antipodarum*) [88], black turban snail (*Chlorostoma funebralis*), emarginated dog-whelk (*Nucella ostrina*), oyster drill (*Urosalpinx cinerea*), American star snail (*Lithopoma americanum*), and smooth Atlantic tegula (*Tegula fasciatus*) [30–32], which suggests that acute toxicity due to these pharmaceuticals is very limited, even at higher concentrations. However, with regard to the body mass of the exposed snails, the influence of citalopram became obvious. Snails exposed to citalopram were lighter than control animals. This reduced weight of exposed snails could be a disadvantage, compared to non-exposed conspecifics, resulting in the lower fitness of exposed animals. Decreases in weight or growth due to exposure to an antidepressant has already been observed in other invertebrate species. In *Hylella azteca* and *Daphnia magna*, growth was decreased after exposure to 31 µg/L fluoxetine [89]. Similarly, exposure to fluoxetine, citalopram, and venlafaxine resulted in decreased weight and length of fish [74,75,90]. For humans, weight loss is also a frequent side-effect of both antidepressants [91,92]. However, in two invertebrate species–marbled crayfish and New Zealand mud snail–no effects on weight or growth were detected after exposure to lower concentrations of citalopram (1 µg/L) and fluoxetine (100 µg/L), respectively [29,88].

Behavioural alterations were only assessed in the experiment with venlafaxine, where venlafaxine-exposed snails showed a significantly increased sole detachment from the aquarium walls and bottom when exposed to at least 100 µg/L of the antidepressant. Similar effects have also been observed in other freshwater and marine gastropod species, such as *Leptoxis carinata*, *Stagnicola elodes*, *Tegula fasciatus*, *Urospalpinx cinerea*, *Nucella ostrina*, and *Lithopoma americanum* after exposure to either venlafaxine, citalopram, or other antidepressants [31,32,93]. In gastropod species, it is well-known that locomotion and mucus secretion are strongly serotonin-dependent [94–97]. As venlafaxine acts as a serotonin re-uptake inhibitor and, thus, interferes with the serotonin metabolism, this interaction may disturb the co-ordination of mucus secretion, ciliate epithelium activity, and muscle cell functions [96]. Consequently, snails that are unable to attach to the surface are, most likely, at a higher risk of predation and are unable to feed [31,32]. As a result, "sole detachment" can reasonably be regarded a population-relevant proxy for viability, as behavioural endpoints are often crucial for individual fitness [38,39].

With regard to the investigated biochemical biomarkers, no significant differences between antidepressant-exposed snails and the controls were obvious. Exposure to the antidepressants did not affect the B-esterase activity in big ramshorn snails. In another invertebrate species, *Gammarus locusta*, no effect of the antidepressant sertraline on the activity of AChE could be detected either [98]. Similarly, in juvenile meagre and larval and juvenile brown trout, no evidence for any changes in the AChE activity of fish exposed to either venlafaxine or citalopram was found [24,60]. However, in the invertebrate species Manila clam (*Venerupis philippinarum*) and *Daphnia magna*, as well as in the fish species common goby (*Pomatoschistus microps*), decreases in AChE activity have been reported after exposure to 5 mg/L fluoxetine and 1 mg/L citalopram [26,68]. In contrast to this, Xie, et al. [55] observed an increase in AChE activity in goldfish exposed to 5 µg/L of the antidepressant sertraline.

Similarly, no proteotoxic effects in response to citalopram or venlafaxine were detected in the ramshorn snails investigated in the present study. Likewise, no proteotoxicity was found in juvenile brown trout exposed to the same antidepressants [60]. In contrast, Maulvault, et al. [24] reported an increase in the Hsp70 level in meagre exposed to venlafaxine.

Although snails exposed to venlafaxine or citalopram showed an overall higher superoxide dismutase activity compared to the respective controls, these differences were not significant; which, however, might be due to the relatively small sample size. An impact on the SOD activity in big ramshorn snail can, therefore, not be entirely excluded. In daphnids exposed to citalopram [26], goldfish exposed to sertraline [55], and meagre and brown trout exposed to venlafaxine [24,60], increased SOD activities have been detected. Furthermore, in zebrafish, an increase in SOD activity at 100 ng/L and a decrease at 1–1000 µg/L amitriptyline has been observed [59]. However, in the common goby and *Gammarus locusta,* no effects on SOD activity were found after exposure to 100 µg/L fluoxetine and 1 µg/L sertraline, respectively [68,98].

No significant deterioration of the health status of the hepatopancreas was observed in snails exposed to venlafaxine. Neither the semi-quantitative nor the qualitative assessment revealed an effect of venlafaxine on the tissue integrity of this organ. In humans, injuries of the equivalent organ–the liver–have been described as a side-effect of venlafaxine [99,100]. In contrast, in fathead minnow exposed to venlafaxine and fluoxetine, no histological changes in the liver were detected [27]. In addition, brown trout exposed to venlafaxine did not show any adverse histopathological liver alterations in a previous experiment [60].

In contrast to venlafaxine, moderate histopathological alteration after exposure to 1000 µg/L citalopram was detected in the qualitative assessment. However, for a valid statement on the effect of citalopram, with respect to the histopathology of the hepatopancreas, further investigation with a higher sample size would be necessary. Yet, no studies exist in literature regarding histopathology in invertebrates and antidepressants. However, effects similar to those found in the present study, in response to citalopram, have been detected in *Marisa cornuarietis* exposed to platinum, copper, and lithium, as well as in *P. corneus* exposed to the metformin transformation product guanylurea [42–44]. Furthermore, irregular compartmentation of digestive cells was been reported for *Xeropicta derbentina, Cernuella virgata,* and *Theba pisana* after heat stress, as well as in *Lymnea stagnalis* exposed to the endosulfan-based pesticide Thiodan® [45,101]. In the invertebrate gastropod *Potamopyrgus antipodarum,* a poor physiological status after exposure to the antidepressant fluoxetine has been reported [88]. Moreover, in a former experiment in brown trout, exposure to citalopram led to impaired liver health condition [60]. In addition, in human patients and in rats treated with citalopram, adverse alterations in the liver have also been reported [102–105]. Histopathological impairment of vital organs can, therefore, affect the individual health of an organism and, as a result, have effects at the population level [49–51].

## 5. Conclusions

A decrease in weight and moderate histopathological effects in the hepatopancreas were observed in snails exposed to 1000 µg/L citalopram, whereas venlafaxine induced behavioural effects after 100 µg/L exposure, at a concentration an order of magnitude lower than the LOEC for citalopram. Although the measured environmental concentrations of these antidepressants are in the range of 0.2 µg/L, thus being far lower than the observed LOECs in this study, a possible contribution of the two antidepressants to mixed toxicity, exerted in concert with other antidepressants, should not be disregarded. In addition, the published PNEC for venlafaxine (91 ng/L) [106] is much lower and below the MECs, such that venlafaxine may pose an environmental risk. In order to enable the derivation of an EQS, the EU, therefore, incorporated venlafaxine into the Surface Water Watch List in August 2020 [107,108], a list of potential pollutants that should be carefully monitored in surface water by the EU Member States.

**Supplementary Materials:** The following are available online at www.mdpi.com/article/10.3390/w13131722/s1, Table S1: Detailed overview on the symptoms identified during the semi-quantitative histological evaluation. The experiment with big ramshorn

snail with the respective treatment are listed in rows and the assessed symptoms are listed in columns. Each symptom is divided into three severity categories (white/0: not detected, light grey/0.5: detected in moderate frequen-cy/severity, dark grey/1: detected in high frequen-cy/severity). Values depict the absolute number of samples showing the symptom. CIT=citalopram, VEN=venlafaxine., Table S2: Water quality parameters during the experiment with big ramshorn snail exposed to citalopram. Mean values are shown., Table S3: Water quality parameters during the experiment with big ramshorn snail exposed to venlafaxine. Mean values are shown., Table S4: Operating parameters of the triple quadrupole MS (Agilent 6490 QqQ) in positive mode., Table S5: Specific measurement parameters for venlafaxine and citalopram with LC-QqQ in water samples of the snail experiments. Intraday variations (RSD) is calculated with 1 μg/L standard (10 μl in-jection volume and 4 replicates (n)). Limit of quantification= LOQ., Table S6: Citalopram water concentrations during the experiment with big ramshorn sail exposed to cital-opram. Concentrations are shown in μg/L. LOD= Limit of detection, a.w.e.= after water exchange., Table S7: Venlafaxine water concentrations during the experiment with big ramshorn sail exposed to ven-lafaxine. Concentrations are shown in μg/L. LOD=Limit of detection, b.w.e.= before water ex-change, a.w.e.= after water exchange., Table S8: CRED-criteria of the experiment with big ramshorn snail exposed to citalopram and venlafaxine., Table S9: Detailed statistical information of the assessed parameters of both experiments.

**Author Contributions:** Conceptualization, M.Z. and R.T.; methodology, M.Z., H.E. and S.T.; soft-ware, M.Z.; validation, M.Z., and R.T.; formal analysis, M.Z.; investigation, M.Z., S.T. and H.E.; resources, H.-R.K., C.Z. and R.T.; data curation, M.Z.; writing—original draft preparation, M.Z.; writing—review and editing, M.Z.; visualization, M.Z.; supervision, R.T., C.Z., and H.-R.K.; project administration, R.T. and C.Z.; funding acquisition, R.T. and C.Z. All authors have read and agreed to the published version of the manuscript.

**Funding:** This research was funded by the Ministry of Science, Research and Arts of Ba-den-Württemberg within the Water Research Network Baden-Württemberg (Wassernetzwerk Ba-den-Württemberg), in which the project Effect-Net (Effect Network in Water Research) was em-bedded Grant No. 33-5733-25-11t32/2). Furthermore, the Authors were supported by the Open Access Publishing Fund of the University of Tübingen.

**Institutional Review Board Statement:** Not applicable.

**Informed Consent Statement:** Not applicable.

**Data Availability Statement:** Data are publicly available on https://effectnet-seek.bioquant.uni-heidelberg.de/investigations/9 (21 June 2021).

**Acknowledgments:** The authors thank the Ministry of Science, Research and Arts of Ba-den-Württemberg for funding the Effect-Net project (Effect Network in Water Research), which is part of the Water Research Network Baden-Württemberg (Wassernetzwerk Baden-Württemberg), and are indebted to Thomas Braunbeck for coordinating this project. We acknowledge the support of the Open Access Publishing Fund of University of Tübingen. Furthermore, the authors would like to thank Helene Heyer, Joanna Probst, and Janina Vanhöfen who completed their theses within the project. Further thanks go to Stefanie Jacob, Stefanie Krais, Elisabeth May, Katharina Peschke, Hannah Schmieg, Mona Schweizer, and Sabrina Wilhelm for laboratory support and technical as-sistance and to Stefanie Dietz and Mona Schweizer for constructive comments on the manuscript.

**Conflicts of Interest:** The authors declare no conflict of interest.

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
