# Peer review of "Effects of the Antidepressants Citalopram and Venlafaxine on the Big Ramshorn Snail (Planorbarius corneus)"

_water, doi:10.3390/w13131722_

Round 1

Reviewer 1 Report

Water-1167583

Recommendation: Reject

Full Title: Effects of the antidepressants citalopram and venlafaxine on the big Ramshorn snail (Planorbarius corneus)

Comments to Author

The manuscript by Ziegler et al. entitled Effects of the antidepressants citalopram and venlafaxine on the big Ramshorn snail (Planorbarius corneus) examines the organism level and biochemical effects of two antidepressants in the big Ramshorn snail. Chronic exposures of either compound resulted in minimal effects in snails.

While the research subject fits with the overall journal focus, the manuscript requires significant restructuring, grammatical corrections, and contextualization of results before its ready for publication. Overall the entire manuscript requires adjustments to enhance flow, transitions between paragraphs, and increase the clarity of the overarching research goal. Several grammatical errors need to be fixed (e.g., typos, punctuation). Proper paragrah structure is lacking while multiple sentences can be combine and/or removed to improve clarity and enhance flow. Examples of such are not provided due to a lack in line numbers; however, they can be found throughout the manuscript.

The number of experimental replicates (i.e., experimental chambers per concentration) is unclear. Individual snails were (n=5-7) used as replicates in single experimental chambers rather than using three separate experimental chambers (N=3) containing multiple organisms, which increase the statistical power. It is also unclear often water chemistry recorded, similar to the number of samples taken throughout the 29 day study to analytically verify water concentrations. These are standard toxicity testing procedures which the author failed to describe and is particularly concerning due to the differences in mean pH conditions during each study. Both antidepressants are weak bases with pKa values around 10.0 and can significantly influence the uptake and corresponding toxicity of each chemical.

The overall manuscript requires significant alterations in flow an structure. Results are reported in the methods section. The results seem to be also copy and pasted in the discussion, almost word for word in certain instances. Within the discussion too much discussion focuses around insignificant trends and thus results are stretched. The author minimally contextualizes the results within the greater scientific topic of pharmaceutical ecological risk assessment.

Minor Comments

Introduction states endpoints but minimally rationalizes the importance of each for examining the adverse effects of antidepressants in the Ramshorn snail. The results presented and those previously reported are minimally contextualized to demonstrate their relevance for the ecological risk assessment of antidepressants and/or pharmaceuticals.

The description of the water filtration needs to be moved up to where it is first mentioned in the manuscript.

How does the author know snails were exposed to a constant antidepressant dose over 29 days when 50% water changes were only conducted once per week?

How often were water samples taken for analytical verification of the exposure water concentration? How often were basic water chemistry parameters measured in exposure tanks? Additionally, there’s no mention regarding how often mortality was assessed.

Results of the behavior analysis are presented in the methods section instead of the results.

Gonad histological analysis and reproductive status were not mentioned in the original objectives for the study.

It is inappropriate to state venlafaxine decreased growth (i.e., body weight) when there’s no statistical difference.

Author Response

Answer: We attached the revised wordfile with track changes. We added line numbers in the revised version. Furthermore, there was a hyperlink error and the results section was copied into the discussion. We addressed this. Sorry for the inconvenience.

Introduction states endpoints but minimally rationalizes the importance of each for examining the adverse effects of antidepressants in the Ramshorn snail. The results presented and those previously reported are minimally contextualized to demonstrate their relevance for the ecological risk assessment of antidepressants and/or pharmaceuticals.

Answer: We addressed the population relevance of the assessed endpoints.

The description of the water filtration needs to be moved up to where it is first mentioned in the manuscript.

Answer: We added this.

How does the author know snails were exposed to a constant antidepressant dose over 29 days when 50% water changes were only conducted once per week?

Answer: We added information on this in the methods. We measured water concentrations at the beginning, after 2 weeks before and after water exchange and at the end of the experiment. The concentrations were comparable before and after water exchange.

How often were water samples taken for analytical verification of the exposure water concentration? How often were basic water chemistry parameters measured in exposure tanks? Additionally, there’s no mention regarding how often mortality was assessed.

Answer: We added information on the water sampling and the measurement of the water parameters. And we added the assessment of the mortality on a daily basis.

Results of the behavior analysis are presented in the methods section instead of the results.

Answer: In the method section the detached snails from the venlafaxine experiment are mentioned to constitute why this endpoint was only assessed in the venlafaxine experiment. Furthermore, in the methods the assessment was explained in detail because this behaviour did not occur from the beginning of the experiment. We changed this in the behaviour section in the methods. Final results are presented in the section results.

Gonad histological analysis and reproductive status were not mentioned in the original objectives for the study.

Answer: The gonad histology is mentioned in the method section and was only assessed to unravel histopathological effects due to the chemical or reproduction status. However, no difference in the reproduction status occurred.

It is inappropriate to state venlafaxine decreased growth (i.e., body weight) when there’s no statistical difference.

Answer: We addressed this and deleted it in the discussion.

Reviewer 2 Report

In the manuscript water-1167583 entitled “Effects of the antidepressants citalopram and venlafaxine on the big ramshorn snail (Planorbarius corneus)” the authors conducted experiments with P. corneus exposed to higher concentrations (1 – 1000 µg/L) of the antidepressants citalopram and venlafaxine. Authors investigated apical endpoints, like weight and mortality, behavioral changes, biochemical biomarkers B-esterase activity, Hsp70 stress protein level and superoxide dismutase activity, as well as the tissue integrity of the hepatopancreas in the exposed snails.

The MS is really well written and provides new and original data, considering that depression is a huge health issue and therefore, treatment of depression has also become a topic of high concern.

The introduction briefly describes the current state of the research the study in a broad context and define the purpose of the work and its significance, including specific hypotheses being tested. Results are well presented. Discussion is well written. Conclusions are supported by the results.

While I enjoyed the flow of the paper, I could not overcome the sense that there are some minor issues that could addressed to improve the quality of the manuscript prior to publication in Water. Also, I recommended to add line numbers in the revised version.

Specific comments

  • The reason why they selected 0, 1, 10, 100 and 1000 µg/L concentration is not provided. Please, add an explanation.
  • Once a week, 50 % of the test solution was renewed. Why did the authors choose this time? Please, explain.
  • It is not clear the age of the snails. Please, add.
  • Results of analytical venlafaxine and citalopram water concentrations should be provided in the results.
  • Figure 1. Caption should be provided as text. Please, add.
  • References should be rewritten following the authors’ guidelines.

Author Response

Answer: We attached the revised wordfile with track changes. We added line numbers in the revised version. Sorry for the inconvenience.

The reason why they selected 0, 1, 10, 100 and 1000 µg/L concentration is not provided. Please, add an explanation.

Answer: We added this in the introduction: the lowest concentration was environmentally relevant. Concentrations were the same as in a former experiment due to comparability as stated in the Experimental design section.

Once a week, 50 % of the test solution was renewed. Why did the authors choose this time? Please, explain.

Answer: We added a statement on this in the Method section

It is not clear the age of the snails. Please, add.

Answer: We added information on this in the methods.

Results of analytical venlafaxine and citalopram water concentrations should be provided in the results.

Answer: results are provided in the table as “aqueous citalopram/venlafaxine concentration (µg/L)”. We added detailed information on the water concentrations in the supplement.

Figure 1. Caption should be provided as text. Please, add.

Answer: We provided the caption as text. This was formatted by the journal.

References should be rewritten following the authors’ guidelines.

Answer: We addressed this. The final formatting of the references is presumably done by the journal.

Round 2

Reviewer 1 Report

Line 73-74: The clarifying comment added skips a level of biological organization, going from organ to population level effects, rather than demonstrating how histological effects impact the organism, which in turn could potentially have X level of effect on the population. Histological effects at the organ level need to be linked to effects at the individual organism level before impacts at the population level can be rationalized, consistent with the adverse outcome pathway conceptual framework linking molecular initiating events (i.e., binding of a chemical to a biological target) to an adverse outcome at the individual level that ultimately leads to impacts at the population level.

Line 141: Please clarify what is meant by "extensive pollution".

Line 155-156: This phrase doesn't clarify which water quality parameters were measured at the beginning and end of the experiment. Additionally, water quality parameters must be measured daily per standard national and international toxicity testing guidelines to prove experimental conditions (i.e., water quality) are acceptable (e.g., > 60% dissolved oxygen saturation, pH between 6-9, constant water temperature varying less than 2 degrees Celsius).

Line 265: Clarify the error associated with each mean value.

Line 455-456: Great. However, be more concise. You haven't demonstrated that the present histological effects observed impact the organism level an are plausibly linked to population level impacts. Have previous researchers published any data that you could use, especially with invertebrates rather than fish (i.e., citation 46)?

Previous comments insufficiently addressed:

Introduction states endpoints but minimally rationalizes the importance of each for examining the adverse effects of antidepressants in the Ramshorn snail. The results presented and those previously reported are minimally contextualized to demonstrate their relevance for the ecological risk assessment of antidepressants and/or pharmaceuticals.

Gonad histological analysis and reproductive status were not mentioned in the original objectives for the study. Even if it was an ad hoc analysis the topic should be introduced in the introduction to portray a succinct appropriately planned study with a specific research question. 

Author Response

Answer: Thank you for your detailed revision of the manuscript. We addressed your comments and also send the manuscript to the MDPI english check for extensive english editing, which was then implemented into the manuscript.

Line 73-74: The clarifying comment added skips a level of biological organization, going from organ to population level effects, rather than demonstrating how histological effects impact the organism, which in turn could potentially have X level of effect on the population. Histological effects at the organ level need to be linked to effects at the individual organism level before impacts at the population level can be rationalized, consistent with the adverse outcome pathway conceptual framework linking molecular initiating events (i.e., binding of a chemical to a biological target) to an adverse outcome at the individual level that ultimately leads to impacts at the population level.

Answer: We rewrote this. As stated in the sentence before, effects on the hepatopancreas affect the general health and therefore the health status of an individual which in return can have an effect on population level.

Line 141: Please clarify what is meant by "extensive pollution".

Answer: We changed this.

Line 155-156: This phrase doesn't clarify which water quality parameters were measured at the beginning and end of the experiment. Additionally, water quality parameters must be measured daily per standard national and international toxicity testing guidelines to prove experimental conditions (i.e., water quality) are acceptable (e.g., > 60% dissolved oxygen saturation, pH between 6-9, constant water temperature varying less than 2 degrees Celsius).

Answer: We added this. A daily measurement of the water parameters was not carried out. However, the parameters were comparable at the beginning and the end of the exposure.

Line 265: Clarify the error associated with each mean value.

Answer: We clarified this in the method in the statistics section.

Line 455-456: Great. However, be more concise. You haven't demonstrated that the present histological effects observed impact the organism level an are plausibly linked to population level impacts. Have previous researchers published any data that you could use, especially with invertebrates rather than fish (i.e., citation 46)?

Answer: We added a statement to clarify, that for a valid statement more data would be necessary. As stated before yet no studies on histopathology in invertebrates and antidepressants exist. However, for fish histopathological alterations can be linked to population levels (Schwaiger et al 2001 (ref 46)) and in snail species harbour contaminants causing histopathological alterations could also be linked to population levels (Bighiu et al 2017 (ref 47)) as well as in invertebrate crab species (Stentiford et al 2005 (ref 48)), which was added in the manuscript.

Previous comments insufficiently addressed:

Introduction states endpoints but minimally rationalizes the importance of each for examining the adverse effects of antidepressants in the Ramshorn snail. The results presented and those previously reported are minimally contextualized to demonstrate their relevance for the ecological risk assessment of antidepressants and/or pharmaceuticals.

Answer: We added information on this.

In each section of the introduction the reason why the biomarker was used is stated now: Behaviour: for antidepressants it was previously shown that behaviour was affected also in invertebrates. Histopathology: the midgut gland is responsible for the elimination of chemicals, such as antidepressants. Stressproteins: already published data by Maulvault et al indicate that antidepressants can have an impact on stressproteins in fish. SOD: similarly, effects of antidepressants on oxidative stress markers such as SOD are published. B-Esterase: Likewise, effects of antidepressants on the B-esterase acetylcholinesterase are available in literature. These effects already published could indicate that both antidepressants also have similar effects in big ramshorn snail.

Now, in the discussion the environmental relevance is stated for each effect: Weight: a reduced weight could result in a disadvantage compared to conspecifics. Behaviour: the shown behavioural changes could result in higher predation risk or an inability to feed which can in result have an effect on population level. Histopathology: histopathological changes were shown to affect individual health and, therefore, have an impact on population health in fish and invertebrates.

Gonad histological analysis and reproductive status were not mentioned in the original objectives for the study. Even if it was an ad hoc analysis the topic should be introduced in the introduction to portray a succinct appropriately planned study with a specific research question. 

Answer: We added a statement in the introduction.
